# Aluminum Adjuvants—‘Back to the Future’

**DOI:** 10.3390/pharmaceutics15071884

**Published:** 2023-07-04

**Authors:** Donatello Laera, Harm HogenEsch, Derek T. O’Hagan

**Affiliations:** 1Technical Research & Development, Drug Product, GSK, 53100 Siena, Italy; donato.1982@libero.it; 2Global Manufacturing Division, Corporate Industrial Analytics, Chiesi Pharmaceuticals, 43122 Parma, Italy; 3Department of Comparative Pathobiology, College of Veterinary Medicine, Purdue University, West Lafayette, IN 47906, USA; 4Research & Development, GSK, Rockville, MD 20850, USA

**Keywords:** aluminum hydroxide, aluminum phosphate, antigen adsorption, multivalent vaccines, adjuvant combinations, Toll-like receptors, analytical characterization, particle size

## Abstract

Aluminum-based adjuvants will continue to be a key component of currently approved and next generation vaccines, including important combination vaccines. The widespread use of aluminum adjuvants is due to their excellent safety profile, which has been established through the use of hundreds of millions of doses in humans over many years. In addition, they are inexpensive, readily available, and are well known and generally accepted by regulatory agencies. Moreover, they offer a very flexible platform, to which many vaccine components can be adsorbed, enabling the preparation of liquid formulations, which typically have a long shelf life under refrigerated conditions. Nevertheless, despite their extensive use, they are perceived as relatively ‘weak’ vaccine adjuvants. Hence, there have been many attempts to improve their performance, which typically involves co-delivery of immune potentiators, including Toll-like receptor (TLR) agonists. This approach has allowed for the development of improved aluminum adjuvants for inclusion in licensed vaccines against HPV, HBV, and COVID-19, with others likely to follow. This review summarizes the various aluminum salts that are used in vaccines and highlights how they are prepared. We focus on the analytical challenges that remain to allowing the creation of well-characterized formulations, particularly those involving multiple antigens. In addition, we highlight how aluminum is being used to create the next generation of improved adjuvants through the adsorption and delivery of various TLR agonists.

## 1. Introduction

New-generation vaccines are being increasingly prepared with highly purified antigens, which improves their safety and tolerability, whilst enabling more simple characterization, but also typically results in decreased immunogenicity. Consequently, adjuvants are added to these vaccines, particularly those comprising recombinant protein subunits, to enhance their ability to induce robust immune responses. Although several new adjuvants have been introduced for specific vaccines over the past 20 years, aluminum adjuvants remain the most commonly used approach, even though they were first discovered nearly 100 years ago [1]. The two main types of aluminum adjuvants included in licensed human vaccines are aluminum hydroxide (AH) and aluminum phosphate (AP). The term ‘alum’ is often incorrectly used to refer to aluminum-containing adjuvants but can be an acceptable abbreviation if associated with a clarifying statement. Alum is a chemical solution of potassium aluminum sulfate, [KAl(SO_4_)_2_·12H_2_O], which is not used as a vaccine adjuvant. While the term ‘alum’ may be used for reasons of simplicity, it is important to define whether this refers to AH or AP, as these adjuvants have very different physical and chemical properties [2]. In addition to AH and AP, a product called Imject^TM^ Alum (ThermoFisher Scientific, Waltham, MA, USA) is sometimes used in preclinical studies. However, Imject^TM^ Alum comprises amorphous aluminum hydroxycarbonate and crystalline magnesium hydroxide [3]. Although it has immunostimulatory activity, it appears to be less potent than AH [4] and is not used in licensed human vaccines. We have previously highlighted that this material should not be used in studies claiming to develop vaccines for human use [5].

The molecular and physical nature of materials used in vaccine adjuvants is very diverse, often for historical reasons. In addition to aluminum salts, adjuvants include oil-in-water emulsions, liposomes, and various natural products, including saponins and Toll-like receptor (TLR) agonists. Consequently, the mechanisms by which these components enhance immune responses are very heterogeneous and often not completely understood, as discussed comprehensively elsewhere [6,7,8]. In general, adjuvants act only at the site of injection and the draining lymph nodes, while systemic effects should be minimized to reduce the potential for poor tolerability. Adjuvants typically increase the recruitment and activation of innate immune cells, enhance antigen uptake or processing and presentation, increase accumulation of lymphocytes in draining lymph nodes, and can promote the development and persistence of germinal centers, which are necessary for the development of a robust immune response [8]. Each individual adjuvant can contribute through some or all of these mechanisms, to a greater or lesser extent. The enhancement of the immune response by adjuvants typically requires coadministration of antigen and adjuvant, although physical association between adjuvants and antigens is often not required. Nevertheless, adsorption of antigens is particularly important for aluminum adjuvants. Hence, the adsorption of antigen to aluminum adjuvants requires the use of specialized analytical techniques, as discussed in this review. The choice of adjuvant in vaccine formulations is determined by multifactorial considerations and is highly dependent on the targeted disease, the chemical and physical nature of the vaccine antigens, and the population for which the vaccine is intended. We have previously argued that, once the need for an adjuvant is concluded, aluminum adjuvants should be the first choice, given their long history of safe and effective use in man, in addition to their wide availability, low cost, and extensive experience of licensure of aluminum-adsorbed vaccines through various regulatory agencies [5]. However, inadequate performance or incompatibility with vaccine antigens when using aluminum adjuvants can necessitate the use of alternative adjuvants such as emulsions or combination adjuvants with TLR agonists. 

In this review, we describe the preparation, analytical techniques, and functional aspects of aluminum adjuvants, followed by a discussion of some recently developed physical methods for assessing the structure of adsorbed antigens. We then describe recent studies in which aluminum adjuvants are used for the development of new and more potent adjuvants in combination with TLR agonists. 

## 2. The Types of Aluminum Adjuvants

Aluminum-adjuvanted vaccines have traditionally been prepared using two methods. The original historical method arose from the use of a solution of alum, potassium aluminum sulfate [KAl(SO_4_)_2_·12H_2_O], to purify tetanus and diphtheria toxoids by precipitation of the proteins. It was observed that the alum-precipitated antigen was more immunogenic than the toxoids alone upon injection in guinea pigs [1], which triggered the original use of aluminum salts as adjuvants. However, the composition and physical properties of formulations prepared using this approach depended very much on the buffer used for the antigen preparation and the precipitation conditions [9,10], which often resulted in formulation variability and inconsistency. Antigens are commonly prepared in phosphate buffers, in which case the alum precipitates are described more accurately as aluminum hydroxyphosphate sulfate [9]. These salts are similar to aluminum phosphate adjuvants (see below); consist of aggregates of platy nanoparticles, as seen under transmission electron microscopy; and have an amorphous X-ray diffraction pattern [9]. However, the original approach of the precipitation of antigens in the presence of alum has now largely been superseded by the adsorption of vaccine antigens onto preformed aluminum hydroxide or aluminum phosphate adjuvants. This method allows for the consistent production of well-defined salts and the full characterization of the adjuvants prior to formulation with antigens, which offers better control over the adsorption. The choice of AH versus AP adjuvants in vaccine formulations is largely determined by the nature of the antigens and the requirement for adsorption to enable an optimal immune response. Importantly, the interactions between antigen and adjuvant are significantly affected by the excipients in the final vaccine formulations (Figure 1) and need to be thoroughly evaluated for each vaccine. 

Aluminum hydroxide (AH) adjuvants are prepared by adding sodium hydroxide to a solution of aluminum ions under carefully controlled conditions. Temperature, concentration, and the speed of mixing are among the factors that influence the physicochemical properties of the adjuvant produced [11]. The product is typically a poorly crystalline boehmite, AlOOH, which has a very different structure from crystalline Al(OH)_3_ [12]. Electron microscopy shows that AH adjuvants consist of nanoparticulate fibers that form loose microparticle aggregates. In contrast, gibbsite, a common naturally occurring form of crystalline Al(OH)_3_, which is not used as an adjuvant, consists of hexagonal nanoparticles [13]. Gibbsite has a small surface area compared with boehmite and is much less effective than AH adjuvants in activation of monocytes [13]. The precise conditions under which AH adjuvants are prepared varies between manufacturers and among different products from the same manufacturer, resulting in significant differences in adsorption behavior and physical characteristics [14]. Furthermore, differences in the aluminum solution used to generate AH adjuvant determine the composition of the final product. The most commonly used solutions are aluminum chloride and alum, with the latter resulting in the replacement of some hydroxyls by sulfate. Although it is not clear if these physicochemical differences translate into different immunological outcomes, it is obviously prudent not to switch the source and type of adjuvant during preclinical and clinical studies. 

Aluminum phosphate (AP) adjuvants are prepared by the precipitation of aluminum ions under alkaline conditions in the presence of phosphate. The addition of phosphate ions results in the formation of aluminum hydroxyphosphate, Al(OH)_×_ (PO_4_)_y_, in which a proportion of the hydroxyls are replaced by phosphate. The ratio of hydroxyls to phosphate varies depending on the precipitation conditions [11]. AP adjuvants are amorphous, i.e., non-crystalline, because the incorporation of phosphate interferes with the crystallization process. The ultrastructure of AP adjuvants shows primary spherical nanoparticles, about 50 nm in diameter that typically form loose microparticle aggregates [9]. 

Amorphous aluminum hydroxyphosphate sulfate (AAHS) is an aluminum-containing adjuvant produced by Merck & Company (Rahway, NJ, USA), and is used in several commercial vaccines, including Recombivax, Gardasil, and Vaxelis. The adjuvanted vaccines are prepared by mixing antigens with preformed AAHS in the standard way, or alternatively by using the original method of precipitation of antigens in phosphate buffer with potassium aluminum sulphate (alum). The preformed AAHS has a point of zero charge of about 7, meaning that it does not carry a surface charge at neutral pH. Ultrastructurally, it consists of platy nanoparticles that are similar to AP and alum-precipitated vaccines [15]. Thus, AAHS can be considered a different form of AP adjuvant with a relatively low P:OH ratio in which the presence of sulfate is due to the inclusion of KAl(SO_4_) as a starting material. There is no reason to believe that the biological properties and safety profile of AAHS are different from AP. 

Some licensed vaccines, such as Twinrix and Infanrix Hexa, contain both AH and AP adjuvants. The vaccines are prepared by mixing selected antigens with either AH or AP adjuvant, and then by combination of the optimally adsorbed components. This leads to redistribution of phosphate on the adjuvants, as phosphate desorbs from AP and replaces some hydroxyls of AH [16]. The resulting increase in the point of zero charge of AP and decrease in the point of zero charge of AH may affect the adsorption of vaccine antigens [16]. Electron microscopy shows aggregates of nanofibers adjacent to or occasionally mixed with aggregates of platy nanoparticles indicating that the adjuvants retain their primary particulate structure upon mixing [17,18]. This is consistent with experiments showing that treatment of AH adjuvant with phosphate buffer results in substitution of surface hydroxyls by phosphate without affecting the crystalline structure [19]. 

A detailed list of current FDA-approved aluminum adjuvant-containing vaccines, with the type and dose of aluminum adjuvant, is provided in Table 1. It is noteworthy that AH and AP are present in a similar number of licensed vaccine products and that the final dose ranges from 0.13 to 0.6 mg Al^3+^.

## 3. Physicochemical Characterization of Aluminum Adjuvants

A number of established assays are available that can be used to characterize aluminum adjuvants and to ensure consistency between batches [11]. Structural information can be obtained using X-ray diffraction (for AH adjuvants only), spectroscopy (Fourier transform infrared, nuclear magnetic resonance, Raman), and transmission electron microscopy [11,20]. Here, we review some additional characteristics of aluminum adjuvants that can be assessed and may have functional relevance.

### 3.1. Particle Size

Aluminum adjuvants are typically composed of primary nanoparticles that form irregularly shaped aggregates with dimensions of 1 to 20 μm [11]. The particle size and shape have important implications for the efficiency of particle uptake by immune cells through phagocytosis [21,22,23]. However, the reported size of the aggregates varies widely and depends on the methodology used to measure particle size, the tonicity and pH of the dispersion medium, and any dilution factor. The particle size can be determined using laser diffraction, dynamic light scattering, or microflow imaging. The surface charge of aluminum adjuvants is pH-dependent (see Section 3.2), and larger particle sizes are observed when the pH approaches the point of zero charge as the electrostatic repulsion is decreased [24]. Similarly, the addition of sodium chloride can obscure surface charges and enhance particle aggregation [25]. However, dilution of aluminum adjuvants in saline caused a decrease in particle size [18].

### 3.2. Surface Charge

The aluminum ions at the surface of AH nanoparticles are coordinated with a hydroxyl that can accept or donate a proton depending on the pH of the dispersion medium. As a result, AH has a pH-dependent surface charge. Its point of zero charge (PZC) is 11.4 and it is positively charged at neutral pH [26]. In the case of AP, a proportion of the surface hydroxyls is replaced by phosphate because of the higher affinity of aluminum for phosphate. Commercial AP adjuvants have a P:Al ratio of 1.1–1.15:1 and a PZC of around 5, which gives them a negative surface charge at neutral pH [27]. A greater proportion of surface hydroxyls results in a higher PZC. Thus, AAHS with a P:Al ratio of 0.3 has a higher PZC and is neutral at pH 7. The surface charge can be determined using a Zetasizer instrument.

### 3.3. Surface Area

The primary nanoparticles that make up the aluminum aggregates afford the adjuvants a very large surface area, estimated at 514 m^2^/g for AH adjuvant, based on water adsorption measured using gravimetric FTIR spectroscopy [28]. The surface area can also be determined using the Brunauer–Emmett–Teller (BET) theory following nitrogen adsorption. However, a smaller surface area was reported for AH compared with water adsorption [25], because dehydration of the samples results in agglomeration of particles with loss of surface area. While these methods cannot be used for AP adjuvants, the ultrastructure of AP, composed of 50 nm nanoparticles, suggests that AP also has a very large surface area.

### 3.4. Adsorption

The large surface area of aluminum adjuvants allows for a high adsorptive capacity for antigens, which can be used as a key tool to allow characterization of the adjuvants. Importantly, adsorption of antigens can impact the quality and magnitude of the immune response and may enhance or decrease the stability of antigens [5]. It should be noted that the dose of antigens in vaccine formulations is typically low, and usually well below full adsorptive capacity. Adsorptive capacity is affected by the type of antigens, the buffer (pH, ionic strength, composition), and other excipients, including the presence of stabilizers or surfactants. The major mechanisms involved in adsorption are ligand exchange of phosphates on the antigen with surface hydroxyls on the adjuvants, along with electrostatic and hydrophobic interactions [2]. Since electrostatic mechanisms play an important role in adsorption, the adsorptive capacity of AH is often determined using a protein that is negatively charged at neutral pH, such as bovine serum albumin (BSA) or ovalbumin, while the adsorptive capacity of AP is typically evaluated with a positively charged protein, such as lysozyme [29]. However, recent studies have suggested that ligand exchange contributes to the adsorption of BSA in spite of the fact that only 0.6% of its serine and threonine residues are phosphorylated [30]. Marked differences were observed in the adsorption mechanisms between BSA and two different types of AH as revealed by changing the pH and tonicity of the BSA solution [30].

It has long been recognized that the strength of adsorption of protein antigens to aluminum adjuvants can increase over time [10,31,32], which likely reflects structural changes in the adsorbed antigens as discussed previously [5]. These structural changes may improve the immunogenicity of the vaccine formulation, but can also lead to deamidation and loss of epitopes, as demonstrated for recombinant protective antigen of the anthrax bacillus [33]. The stability and immunogenicity of adsorbed vaccine formulations should be studied over time. 

### 3.5. Elemental Composition 

The presence of impurities in aluminum adjuvants can be determined using inductively coupled plasma mass spectrometry (ICP-MS) [34]. Differences in the type and quantity of metal ions were reported between AH adjuvants obtained from different manufacturers and different batches, which are likely caused by differences in the sources of aluminum salts, chemicals, and water used during the production process [34]. The presence of sulfur likely reflects the use of alum as the starting material as discussed above. Some contaminants such as copper may affect the stability of adsorbed antigens [34].

## 4. Biological Differences between AH and AP Adjuvant

It is often suggested that AH induces more robust immune responses than AP [10], but few published studies have directly compared the potency of vaccines formulated with AH vs. AP. AH induced a stronger immune response against tetanus toxoid, a snake venom, and a viral glycoprotein than AP in mice and guinea pigs [35,36,37]. However, no differences were observed for diphtheria toxoid [35] and for an anthrax recombinant protein [38]. Furthermore, evaluation of Hib-CRM197 conjugate vaccines in infants showed that aluminum adjuvants significantly enhanced the immune response, but there was no difference between AH and AP [39]. In contrast, during preclinical studies on the development of a human papillomavirus (HPV) vaccine, a significantly stronger immune response was observed for HPV virus-like particles formulated with AAHS and AP versus AH [15]. This limited number of observations suggests that AH may induce a stronger immune response in certain vaccine formulations, but this is likely antigen-dependent. 

In vitro studies using simulated interstitial fluid and a rabbit study using isotope-labeled adjuvants indicate that AP dissolves more rapidly than AH following injection [40,41]. This is consistent with the observations that AH persists longer than AP in muscle of nonhuman primates and rats after injection of aluminum-adjuvanted vaccines [42,43]. The shorter retention of AP vs. AH may be a factor in the lower risk of granulomas following vaccination with AP-containing vaccines than AH-containing vaccines [44].

The full mechanism of action of aluminum adjuvants is still poorly understood, as reviewed extensively elsewhere [45,46,47]. Most studies have focused on AH and few experiments have compared the biological effects of AH and AP. Both AH and AP induce the release of IL-1β and IL-18 from mouse dendritic cells and human peripheral blood mononuclear cells in a caspase-1-dependent manner [48,49]. These cytokines may play a role in the recruitment of inflammatory cells to the injection site, although their role in the augmentation of antibody responses by aluminum adjuvants is uncertain. Aluminum adjuvants enhance antigen presentation and T cell activation by dendritic cells [48,50]. AH induced antigen presentation of ovalbumin (OVA) to a greater extent than AP in vitro as determined by the activation of OVA-specific T cells [48]. As OVA adsorbs more strongly to positively charged AH, it is unclear to what extent the increased antigen presentation was due to increased uptake of OVA adsorbed to AH or other physical differences between AH and AP. However, these results are consistent with the stronger upregulation of proteins associated with antigen processing and presentation following treatment of human monocytes with AH compared to AP [51]. Exposure of human monocytic THP-1 cells to AH and AP revealed that AP was more cytotoxic than AH [52]. This may lead to greater release of danger-associated molecules, such as uric acid, ATP, and DNA, which could stimulate a stronger inflammatory response, but assessment of the innate immune cells in muscle after injection of AH and AP indicated a weaker neutrophil response induced using AP and no difference in the percentage of monocytes and macrophages [40].

In summary, few studies have systematically assessed differences in the biological function of AH and AP. Overall, it appears that AP adjuvants dissolve more quickly in vivo and are less likely to induce formation of granulomas. However, the choice of incorporating AH or AP into vaccine formulations to induce an optimal immune response is very much antigen-dependent and must be made empirically.

## 5. Safety of Aluminum-Adjuvanted Vaccines

Aluminum-containing adjuvants have been used for over 80 years in human vaccines, with millions of doses injected annually in infants, adolescents, and adults. Based on this unsurpassed historical record, aluminum adjuvants are regarded as safe and well tolerated [53]. A small proportion of injected individuals may develop granulomas or contact hypersensitivity to aluminum following injection of aluminum-adjuvanted vaccines [54,55]. Aluminum is a widespread element in the environment and is present in food, personal care products, and medications. There is no evidence that injection of aluminum-containing vaccines increases the aluminum levels in blood above baseline or minimum risk levels [56,57] and causes systemic disease or neurologic disorders. The safety of aluminum adjuvants is further supported by epidemiologic studies of patients who receive frequent injections of allergens formulated with AH during the course of subcutaneous immunotherapy (SCIT) for allergic diseases [58]. About two thirds of allergen immunotherapy products used in Europe contain AH [59]. Comparison of patients treated using conventional antiallergy medication (oral antihistamines or intranasal corticosteroids) with patients undergoing SCIT with AH-containing allergens over a 10 year period demonstrated a lower incidence of autoimmune disease, ischemic heart disease and overall mortality in the latter group [60]. SCIT involves injections over a 3-year period with about 100 times more AH than what is present in 3 doses of aluminum-adjuvanted hepatitis B vaccines. 

## 6. Analytical Characterization of Aluminum-Adsorbed Antigens

Aluminum-based adjuvants are insoluble and are used as suspensions comprising opalescent dispersions of micron-sized particles. This means that they remain insoluble in standard aqueous buffers, including those normally used for vaccine formulations. As a consequence, the characterization of adsorbed protein for quality control, including measuring the antigen content, aggregation/degradation profile, and structural conformation, represents a challenging issue, since the turbidity of aluminum suspensions interferes with most conventional assays for protein characterization, such as colorimetric (BCA, Lowry, Bradford), chromatographic (RP-UHPLC, SEC), spectroscopic (FTIR, CD, fluorescence), and electrophoretic (SDS-PAGE/Western blot) approaches. To overcome this issue, scientists have developed procedures to desorb proteins from aluminum adjuvants (using phosphate buffers at high pH, followed by centrifugation steps), or to directly dissolve the protein aluminum complex (in a citrate-based solution), to separate the protein from aluminum particles prior to analytical characterization. However, these approaches have significant limitations, since they are time consuming, laborious, and often inefficient. In addition, they can alter protein structure during the desorption/dissolution step and can result in highly variable, protein-dependent outcomes, which is particularly problematic for multivalent/combination vaccines. 

Efforts in the last two decades to circumvent these issues have resulted in the development of new analytical tools that can avoid the need for protein desorption or adjuvant dissolution, potentially allowing for direct characterization of adsorbed protein(s). In this section, we describe the current state of the art for these approaches based on their capability to evaluate secondary or tertiary protein structures. An important consideration is that most of these approaches can only be applied to monovalent vaccine formulations, since their low specificity makes it challenging to discriminate signals coming from more than one protein. In the last section, we discuss approaches for the evaluation of multivalent vaccine formulations. A list of the available analytical tools discussed is shown in Table 2, along with the type of information that can be acquired, the current status in vaccine evaluation, and the applicability for monovalent versus multivalent vaccines.

### 6.1. Analytical Tools to Assess Secondary Protein Structure

#### 6.1.1. Attenuated Total Reflectance Fourier Transform Infrared (ATR-FTIR) Spectroscopy

FTIR spectroscopy is one of the more widely used vibrational spectroscopic methods that can generate important information on protein conformation, focusing on prediction of β-strands within the secondary structure [61]. However, due to the overlapping of the absorption band of water with those of peptide bonds and the low protein signal relative to the water signal, the implementation of attenuated total reflectance (ATR), which measures the reflected light at a crystal/liquid interface, was necessary to evaluate antigen adsorbed on aluminum. In addition to this technical correction, the interpretation of protein spectra from ATR-FTIR analysis is not always straightforward, with extensive experience necessary for accurate interpretation. To overcome the weakness of FTIR spectroscopy in studies of proteins at low concentration (i.e., aluminum adjuvanted vaccines), a preliminary step of centrifuging AH samples to obtain a more concentrated protein–AH complex was introduced. The results suggested that the interactions between the hydrophilic surfaces of AH gel and the model proteins (in monovalent samples), did not alter the secondary structure of the proteins [62] relative to their solution states. However, previous studies using different model antigens (BSA, lysozyme, and ovalbumin) suggested that protein adsorption onto AH could result in changes in protein conformation, based mainly on thermal stability data [63,64,65]. It was also demonstrated that upon thermal denaturation, the structural transitions between native and denatured states was very similar to what was seen for the proteins adsorbed onto AH adjuvant. This suggested that the thermal stability of proteins is minimally affected by adsorption onto AH adjuvant [66]. Similar findings for human immunodeficiency virus (HIV) glycoprotein 41 (gp41) also support the conclusion that adsorption does not affect the secondary protein structure after adsorption [67]. On the contrary, when FTIR analysis was applied to BSA as a model antigen, the spectral differences between adsorbed and solution state revealed that a modified protein structure was induced following adsorption onto AH [49], which was consistent with previous work [45,46], and highlighted that results are usually antigen dependent. Near-infrared (NIR) spectroscopy has also been used for direct and nondestructive evaluation of BSA adsorbed to AH [50]. The data on NIR absorbance in the range 700–1300 nm correlated well with the UV-vis absorbance at 280 nm of unformulated BSA, and the assay did not seem to be affected by AH sedimentation rate, buffer compositions, or different AH batches.

#### 6.1.2. Circular Dichroism (CD) Spectroscopy

CD has been recognized as a valuable technique for examining the structure of proteins both in solution and following adsorption to aluminum [68]. A CD signal is obtained when a chromophore is chiral (i.e., optically active) in the following situations: (I) its inherent structure, (II) it is covalently linked to a chiral center, or (III) it is placed in an asymmetric environment due to the specific conformational structure adopted by the molecule [69]. In protein antigens, the chromophores of interest are represented by the peptide bonds (absorption below 240 nm), aromatic amino acid side chains (absorption in the range 260 to 320 nm), and disulfide bonds (around 260 nm). In addition, nonproteins can absorb over a wide spectral range (from 300 up to 650 nm). The earliest references for comparing secondary structure of antigens using CD in solution and AH-adsorbed state are from 2012, when two groups reported different conclusions based on different antigens. The first team highlighted substantial changes in the structure of diphtheria toxoid (DT) antigen upon adsorption [68] while the second one demonstrated that the secondary structure and the conformation of the hepatitis B surface antigen (HBsAg) remained intact during and after adsorption onto AH [70]. In both of these studies, the conclusions were confirmed through the application of orthogonal approaches (i.e., FTIR, IF, and EF).

### 6.2. Analytical Tools to Assess Tertiary Protein Structure

#### 6.2.1. Fluorescence Spectroscopy

Fluorescence spectroscopy is probably the technology most widely used to characterize adsorbed proteins, since it can provide useful information on their conformation and tertiary structure. Fluorescence is used to visualize the aromatic amino acids in a primary protein sequence, such as tryptophan (Trp), tyrosine (Tyr), and phenylalanine (Phe), through their intrinsic fluorescence (IF). When excited in the UV region (250 nm for Phe, 275 nm Tyr and 290 nm for Trp), these aromatic amino acids can produce a spectrum with maximum emission in the range of 310–360 nm [71]. Changes in the emission spectrum can be related to alterations of conformation/tertiary structure through signals from amino acid residues that would normally remain shielded in the protein inner core and are inaccessible. The first report of IF for an adsorbed protein suggested that the conformation of mouse monoclonal antibody 383 (MMA383) adsorbed on AH could play an important role in performance, since the differences between the most and the least in vivo active formulations of MMA383–AH correlated with changes in the fluorescence spectroscopic properties of the adsorbed antibody [63]. In another study, it was reported that extensive perturbation of the structure of BSA occurred upon interaction with the surface of an insoluble AH salt in comparison to BSA in solution [64]. Similar results were also obtained using model antigens (lysozyme, ovalbumin, and BSA) and vaccine antigens (recombinant ricin toxin A-chain V76M/Y80A (rRTA) and erythrocyte-binding antigen 175 kDA region II-nonglycosylated (EBA-175 RII-NG) from *Plasmodium falciparum*), which were formulated with AH or AP. The authors reported destabilization for all proteins adsorbed onto the aluminum salts, and that stabilizers were required in the final drug product to reduce this effect [65]. In contrast, the tertiary structure of recombinant protective antigen (rPA83) of *Bacillus anthracis* did not change upon adsorption to AH [72]. The application of IF to determine antigen content was demonstrated for both AH and AP using *Bordetella pertussis* filamentous haemagglutinin (FHA), pertactin (PRN), and fimbriae (FIM) as antigens with good accuracy, reproducibility, and sensitivity [73]. This approach was confirmed and improved with a virus-like particle (VLP) antigen (malaria CSP R21 protein) adsorbed to AH, using a high-throughput format [74].

An alternative application of fluorescence spectroscopy is extrinsic fluorescence (EF), also referred to as differential scanning fluorimetry (DSF), in which a low molecular weight fluorescent ligand (dye) binds to hydrophobic parts of the protein. The temperature at which a protein unfolds is measured by an increase in the fluorescence signal, which reflects better recognition by the dye of the hydrophobic exposed regions [75]. A significant advantage of this approach is that the detection of protein unfolding is independent of the presence or position of aromatic amino acids within the protein. EF was used to gain information about conformational changes of different antigens adsorbed to aluminum adjuvants in a high throughput mode for screening different formulation conditions (buffer, pH, excipients, etc.). Sugars and polyols enhanced the physical stability of all three adjuvanted antigens in a concentration-dependent manner [76]. EF was shown to be the most sensitive technique among approaches used to identify structural changes in diphtheria toxoid following adsorption to AH [68]. Interestingly, despite the presence of 5 Trp residues within the primary sequence, no measurable changes were detected in IF between the adsorbed antigen and the solution state. This data could suggest that protein adsorption resulted in a heterogeneous population with molecules in two conformational states, one unaffected and one perturbed. Considering the huge number of variables at play (antigens with different physicochemical properties, aluminum adjuvant sources, antigen and adjuvant dose/ratios, final pH value, buffer, isotonicity modifier, and excipients), it was not surprising to see different outcomes. However, taking into account the totality of published reports, we can conclude that fluorescence spectroscopy is the technique of choice for identifying changes in the tertiary structure of adsorbed antigens in monovalent vaccines.

#### 6.2.2. Differential Scanning Calorimetry (DSC)

DSC is an analytical approach used to characterize the thermal stability of a protein in its native/tertiary form due to evaluation of the heat exchange associated with thermal denaturation when heated at a constant rate. Proteins exist in equilibrium between native (folded) and denatured (unfolded) conformations, and a higher thermal transition midpoint (Tm) indicates a more stable molecule. DSC measures the enthalpy (∆H) of unfolding that results from heat-induced denaturation. DSC is a powerful technique that elucidates the factors that contribute to the stability of a native protein, which is particularly important in the formulation of drug product candidates [77]. The first use of DSC in AH/AP formulations for adsorbed monovalent proteins described the denaturation of model antigens such as BSA and lysozyme, and the observations were supported by other types of analysis (i.e., FTIR, IF, ITC). In a more detailed study using botulinum neurotoxin as a model antigen adsorbed onto AH, the DSC thermograms acquired at the beginning were very different compared to those obtained after storage for nine weeks at 4 °C. In addition, the differences were much more pronounced for samples stored at 30 °C. The authors concluded that the adsorbed protein experienced some degree of unfolding during storage, which resulted in a greater degree of interaction between the proteins and the adjuvant surface and an increase in resistance to desorption [78]. In contrast, another study showed that a vaccine containing a genetically detoxified pertussis toxin (gdPT) formulated on AH in the presence of different TLR agonists displayed similar thermal stability to gdPT in solution, indicating no major perturbation after aluminum adsorption [79].

#### 6.2.3. Nuclear Magnetic Resonance (NMR) Spectroscopy

NMR spectroscopy is used to obtain information about the structure and dynamics of proteins. This approach provides a map of how the atoms are linked chemically, how close they are in space, and how rapidly they move with respect to each other. Most samples are examined in an aqueous solution, but methods are being developed to also allow for evaluation of solid state samples. There are a few examples of NMR application for resolving the structure of aluminum adjuvant-adsorbed antigens. High-resolution solution-state NMR was used for a protein combined with aluminum phosphate adjuvant [80]. Since the massive particle size of the adjuvant precluded elucidation of the detailed structure of the protein in the adsorbed state, the authors performed NMR evaluation on the desorbed protein, which revealed that it readily refolded following desorption and exhibited native structure. 

A more recent study used solid state NMR methodology to characterize the structural features of L-asparaginase from *E. coli* (ANSII) adsorbed to AH [81]. In this paper, the ANSII antigen was stable for several months, since the spectra obtained from sedimented AH-ANSII post-centrifugation were superimposable with those of the rehydrated freeze-dried ANSII. Hence, the three-dimensional structure of the protein was preserved after adsorption to AH. This work was important for the field since it was performed using a commercial vaccine product (Menjugate)—which contains a small amount of antigen adsorbed to a high dose of adjuvant—in the presence of additional excipients, which could adversely impact the outcome. Hence, the high sensitivity of solid state NMR combined with isotopic labeling methodologies can be considered a key innovation in the characterization of aluminum adjuvant-adsorbed vaccines.

### 6.3. Analytical Tools for Multivalent Vaccine Formulation

As stated above, the presence of more than one antigen in the same aluminum adjuvant containing vaccine formulation can be a challenge from an analytical perspective, potentially preventing discrimination between signals raised from different antigens. In the following paragraphs, we discuss the results obtained using different approaches to overcome this issue.

#### 6.3.1. Imaging and Cytometry

The first attempt to directly visualize antigen (BSA, myoglobin, alpha-casein and recombinant protective antigen (rPA) from *Bacillus anthracis*) adsorbed to AH employed confocal microscopy [82]. Using proteins labeled with different fluorophores, the authors demonstrated that, independent of adsorption strength, both monovalent and combination vaccines rapidly reached a uniform antigen distribution across AH particles. More recently, an improvement in confocal microscopy was possible using a high content fluorescence imaging analysis (HCA) approach, with well-characterized mAbs against recombinant virus-like particles (VLP) of human papillomavirus (HPV) [83]. The antigenicity and the integrity of a defined epitope or two nonoverlapping epitopes on the VLPs in their AH-adsorbed form could be assessed in a quantitative manner for quality control, including real-time or accelerated stability testing of final products. The same group subsequently reported on HCA in a multiplexed assay for in situ quantification and distribution of dual antigens adsorbed on AH adjuvant in the vaccine formulation [84].

As an alternative imaging technique, flow cytometry (FC) can represent an option to assess aluminum adjuvant and the adsorbed antigens. As for confocal microscopy or HCA, fluorescence labeling is essential and can be directly performed on antigens or through antigen-specific antibodies. *Neisseria meningitidis* antigens were quantified in mono- and bivalent formulations with a polyclonal antibody [85]. Interestingly, FC showed the best performance from a quality perspective, since it was useful both in characterization of adjuvanted vaccines with respect to freeze damage [86] and the antigen distribution across aluminum particles [87]. In this recent publication, the authors showed that different formulation strategies and differences in electrostatic adsorption strength can result in different distribution of antigens and a time-dependent maturation effect to reach homogeneous and monodispersed antigen–AH complexes (Figure 2). Imaging flow cytometry is the latest version of an all-in-one platform for the measurement of *Neisseria meningitidis* antigens–AP complexes in vaccine formulations, allowing for the detection of protein aggregates [88].

#### 6.3.2. Raman Spectroscopy and Liquid Chromatography–Mass Spectrometry (LC–MS)

The need to control the quality of vaccines requires the development of new analytical methods because of the high molecular weight of antigens relative to classical pharmaceutical products and the high complexity of combination products. Regulatory requirements for vaccines routinely include tightly controlled manufacturing and batch release processes. Two recent papers represent a tentative approach to extend the application of Raman spectroscopy and LC–MS, respectively, for the identification and quantification of tetanus, diphtheria, and pertussis (DTaP) antigens in aluminum-adjuvanted vaccines [94,95]. Raman spectroscopy demonstrated the potential for the identification and differentiation of complex vaccine products [94]. Raman maps obtained from air-dried samples of combination vaccines containing DtaP antigens were used to build a fingerprint of product-specific Raman signatures. The results highlighted the high specificity and sensitivity of Raman measurements in identifying DTaP vaccine products. These promising data should pave the way for further exploitation of the Raman approach for quality control of multivalent vaccines. 

An LC–MS-based method was developed for the quantification of antigens in a final DTaP vaccine using antigen-specific signature peptides and isotopically labeled standards [95]. Individual antigens in a multivalent AH-adjuvanted DTaP vaccine were identified and specifically quantified using LC–MS, with quantification consistent with the concentrations supplied by the manufacturer. This approach demonstrated the consistency of DTaP vaccines with the overall objective of reducing reliance on animal-based testing for vaccine control and release.

#### 6.3.3. ELISA-Based Approaches

Immuno-based in vitro assays are being increasingly used in the vaccine industry to refine, reduce, and replace (3Rs) the use of in vivo animal testing for potency or immunogenicity [96]. Promising results for aluminum-adsorbed vaccines were initially achieved in 1989, when researchers pioneered the first in vitro approach using a direct ELISA to assess vaccine potency (i.e., antigen quantification) without the need for desorption for inactivated vaccine bovine rhinotracheitis (IBRV), pseudorabies (PRV), and porcine parvovirus (PPV) [89]. An improvement on an ELISA-based application for a vaccine quality control assay was a direct Alhydrogel formulation immunoassay (DAFIA) to accurately, sensitively, and specifically determine *Plasmodium falciparum* apical membrane antigen 1 (AMA1)-C1 content, identity, and integrity adsorbed on AH [90]. An adaptation of the DAFIA incorporated a panel of monoclonal antibodies for qualitative analysis of DT adsorbed to AP in combination vaccines containing DT, tetanus toxoid (TT), and inactivated poliovirus [91]. Comparing the antigenic fingerprints (antigenic quality of the adsorbed protein) of different batches of vaccines allowed for an evaluation of product consistency and any changes in epitope availability after desorption. Changes in epitope profile were observed during adsorption of DT, but these conformational changes were reversible after desorption from AP. The authors suggested that these changes could be caused by preferential orientation of the antigen adsorbed to AP, resulting in the disappearance of certain epitopes. Previously, the same group demonstrated that DT undergoes substantial conformational changes upon adsorption by using spectroscopic techniques, i.e., fluorescence and circular dichroism, at least to AH [68]. 

Although reproducible and specific, ELISA methods require separate analysis of each individual antigen in a combination vaccine. Luminex technology was evaluated to allow simultaneous characterization of acellular pertussis (aP), TT, and DT in an AH-adjuvanted TdaP vaccine [92]. Luminex is a solution phase bead-array immunoassay used for the analysis of multiple analytes within a single assay run [93]. This method represents a significant improvement for the in vitro characterization of complex multivalent AH-adjuvanted vaccines by providing information on multiple antigens on the surface of the adjuvant in a single experiment. The authors suggested potential application for larger vaccine combinations and in the presence of other adjuvants since the method worked in the multiplex configuration with good specificity, accuracy, and linearity. The high sensitivity of this assay is a critical parameter for vaccine characterization since antigens in vaccines are usually formulated at very low concentrations.

## 7. Next Generation Aluminum-Based Adjuvants

### 7.1. Limitations of Aluminum Adjuvants

Aluminum-based adjuvants have contributed significantly to the success of vaccination in the control of infectious diseases. They are effective in enhancing the immune response to pathogens for which protection relies on antibody-mediated immune responses. They are inexpensive and have an excellent record of safety. However, the use of aluminum adjuvants in vaccines has some limitations. Inadvertent exposure of vaccines to freezing is common during the storage and transport of vaccines. This causes physical and chemical alterations to the aluminum adjuvants that negatively affect their ability to enhance immune responses [97,98]. However, these changes may be minimized by the addition of stabilizers or other excipients [99,100]. Aluminum adjuvants may not be suitable for certain vaccine antigens. For example, they do not appear to enhance the immune response to split-virus or whole killed virus influenza antigens [101]. Perhaps the most important limitation of aluminum adjuvants is their inability to support a robust cell-mediated immune response, which is required to induce protection against certain pathogens like *Mycobacterium tuberculosis*. The performance of aluminum adjuvants in combination with protein antigens is generally characterized in mice by the induction of an enhanced Th2 immune response, with IL-4 production and increased antibodies (mainly IgG1 and IgE isotypes) [45]. However, the immunological profiles can be affected by the vaccines evaluated and the mouse strains used [102,103]. Accumulated evidence also highlights the induction of a more balanced Th1/Th2 immune profile in humans, which can be influenced by the type of formulation and the antigen [104,105]. Nevertheless, in general, aluminum-based vaccines induce a relatively weak (Th1) immune response, which does not offer optimal protection against certain intracellular pathogens [106]. Consequently, much effort has been applied to switch the aluminum Th2-based response to a more Th1 response. One approach is represented by nanoalum [107], which has smaller particle size than traditional AH, along with changes in the shape and degree of crystallinity of the material [108]. Studies on nanoalum suggest a stronger and more durable antibody response [107], increased antigen uptake by APCs [109], and a switch from Th2 to Th1/Th17 responses [110,111]. Similar conclusions were obtained for aluminum hydroxyphosphate [51,112]. Another approach is to combine aluminum adjuvants with other immunostimulators, in particular TLR agonists, taking advantage of the extensive adsorptive capacity of aluminum adjuvants [106,113]. Several examples in licensed vaccines and in clinical development are discussed in the next section.

### 7.2. Aluminum–TLR Agonist Combination Adjuvants 

The first TLR agonist and aluminum adjuvant combination included in a licensed vaccine for human use is Adjuvant System 04 (AS04), which comprises 3-O-desacyl-4′-monophosphoryl lipid A (MPL) adsorbed onto aluminum, an adjuvant which is included in two GSK vaccines, Cervarix and Fendrix, used, respectively, to protect against human papillomavirus and hepatitis B virus [114]. MPL is a detoxified version of lipopolysaccharide (LPS), isolated from the Gram-negative bacterium *Salmonella minnesota,* R595 strain, and is an agonist of TLR4 [115]. Detoxification is achieved using successive acid and base treatments, allowing for the production of a molecule with retained immunostimulatory activity, but with significantly reduced toxicity relative to LPS. LPS is composed of three covalently linked regions: lipid A-containing glucosamine disaccharide units that carry long chains of fatty acids; the core oligosaccharide; and O-specific polysaccharide chains, containing repeated oligosaccharide units. MPL displays the same basic structure as the lipid A region of LPS. Due to its hydrophobic nature, MPL forms insoluble particles in aqueous media, ranging in size from 100 to 500 nm. It is available as a lyophilized powder to be resuspended in water before sterile filtration and adsorption to aluminum [116]. Due to the lack of chromophores in its structure for UV detection, MPL can be characterized using reverse-phase chromatography (RP-UHPLC) analysis coupled with an evaporative light scattering detector (ELSD) when dispersed in aqueous solution or after desorption from aluminum [117]. Because of MPL’s structural similarity with LPS, endotoxin evaluations using the Limulus amebocyte lysate (LAL) test generates misleading results. To overcome this issue, the rabbit pyrogenicity test (RPT) can be used to measure endotoxin content [118]. MPL adsorbs efficiently to AH through ligand exchange of the phosphate groups on lipid A and through strong electrostatic interaction since it is a negatively charged molecule [119]. Preclinical studies show that AS04 greatly enhances the production of antibodies and induces high levels of memory cells [120]. These features were confirmed in humans with Fendrix, which demonstrated higher seroprotection rates and a longer-lasting antibody response to the HBV vaccine adjuvanted with AS04 than with aluminum salt alone [121]. Similarly, the inclusion of MPL in an HPV vaccine resulted in enhanced and sustained humoral and cellular immune responses compared to AH alone, with higher antibody levels in humans up to 3.5 years after vaccination, a higher frequency of memory B cells, and cross-protection against infection and/or lesions associated with HPV types that are related to HPV-16 and HPV-18 [122].

A second example of an aluminum–TLR agonist combination adjuvant is represented by AS37, which is a small-molecule immune potentiator (SMIP), agonist of TLR7, adsorbed onto aluminum hydroxide [123]. Using a medicinal chemistry approach, we were able to synthesize low molecular weight compounds (500–600 Da) whose benzonaphthyridine (BZN) pharmacophore represented a new chemical entity (NCE) able to activate TLR7 [124]. Although imidazoquinoline (IMQ) compounds, which were TLR7/8 agonists, including imiquimod, had been studied as adjuvants in preclinical models [125], they did not have the molecular characteristics to allow them to be developed as vaccine adjuvants for use in humans [126]. An important feature of the development of SMIPs was to increase vaccine potency/safety by enhancing retention at the injection site to minimize the potential side effects that would result from a more generalized biodistribution. These requirements were satisfied by designing highly soluble SMIPs through addition of a phosphonate functional group to the BZN scaffold that also allowed for adsorption to AH via ligand exchange without affecting TLR7 agonist activity [124]. Comprehensive studies were performed to demonstrate the preparation and characterization, including adsorption stability, of AS37 [127]. To allow for the evaluation of multiple dose levels of TLR7a, AS37 can either be prepared according to the final target concentrations (standard process) or as an AS37 formulation at the highest dose that can be serially diluted with AH, an approach particularly useful for easy clinical evaluation. We recently demonstrated the use of FC analysis to directly highlight association of SMIP on AH particles through an increase in fluorescence signal (blue trace) with respect to plain AH particles (red trace) (Figure 3). The AS37 sample at high SMIP dose (blue trace) showed a distinct and monodispersed peak, indicating that all particles were delivering a similar amount of TLR7a. To prepare AS37 for delivering a lower SMIP dose, the AS37 sample was diluted with plain AH suspension, respectively, at 1:2 (green trace) and 1:3 (orange trace); the resulting AS37 samples had a relative reduction in fluorescence intensity in accordance with the dilution factor (Figure 3). Studies showed that AS37 is flexible and can be used with many different types of vaccine candidates in preclinical models. Importantly, it can switch the immune response towards Th1 with overall increased immunogenicity of the vaccine, as measured using several criteria, in a range of animal models, to include nonhuman primates [128,129,130,131,132]. Clinical evaluation of AS37 using a MenC-CRM197 glycoconjugate vaccine as model antigen showed that the safety and reactogenicity profile was comparable to an AH-formulated vaccine alone but could be dose dependent [133]. Furthermore, a systems biology evaluation in this trial showed an expected increased expression of IFN-mediated genes [134].

An alternative SMIP approach is represented by 3M-052, a synthetic IMQ TLR7/8 agonist with 18-C fatty acyl chain incorporated to enhance hydrophobicity, reduce systemic dissemination, and improve retention at the injection site [135]. However, due to insolubility issues of the compound, this formulation did not have a clear path forward to development. However, 3M-052 was also formulated with AH or entrapped in polymeric nanoparticles and, with an HIV-1 clade C-derived gp140 immunogen (Env), induced robust and durable Env-specific long-lived plasma cells significantly better than AH alone. Since the aluminum-based 3M-052 formulation had an easier path to human testing, following extensive evaluations in NHP [136], these results paved the way for a phase 1 clinical trial to assess the adjuvant potential of 3M-052 in the context of HIV Env antigens. Surprisingly, when the same molecule was used as adjuvant for a SARS-CoV-2 receptor-binding domain (RBD) nanoparticle, 3M-052 alone induced superior systemic and mucosal antibody responses to the 3M-052-AH-adjuvanted formulation [137]. Hence, the performance of this adjuvant may be somewhat antigen-dependent. In contrast to AS37, this approach is complicated by the need to create a nanoparticle dispersion of the insoluble 3M-052 adjuvant to allow for adsorption to AH [138]. An alternative approach, 3M-052-AF, which included DSPG as a helper lipid to allow formation of a nanosuspension (<200 nm), was described and could be adsorbed to AH via ligand exchange [138].

A different approach to the rational design and development of an aluminum-based adjuvant containing an agonist of TLR7/8 was based on the screening of three different IMQ and three oxoadenine (OA) compounds, which were synthesized and evaluated for their TLR7/8 selectivity and potency. The IMQ named UM-3001 showed the greatest potency for human TLR7/8, and the ability to induce, in vitro, both TNF secretion and IFN-γ polarization in newborn cord blood. This IMQ was further engineered using a phospholipid and 3 PEG linkers to create a new IMQ called UM-3003 [139]. In contrast to the highly soluble TLR7a used in AS37, the phospholipidated UM-3003 resulted in a suspension of nanosized lipid particles, which were later adsorbed to AH via ligand exchange. This approach necessitates an additional manufacturing step that needs to be controlled for UM-3003 vs AS37, which is similar for 3M-052-AF, before formulation onto aluminum. The UM-3003-AH adjuvant was tested in a neonatal mouse model and resulted in a balanced Th1/Th17-polarized cell-mediated and IgG2c-skewed humoral response to a licensed acellular vaccine (Infanrix). AS37 has also previously been evaluated using an established aP vaccine and demonstrated enhanced Th1/Th17 responses, along with enhanced protective immunity in an adult murine model [140]. Overall, these data highlighted aluminum-TLR7/8a as a promising adjuvant that may enable an improved vaccine against pertussis and other pathogens. The most advanced TLR7/8 agonist in terms of clinical development is represented by a whole virion-inactivated SARS-CoV-2 vaccine (BBV152) adjuvanted with AH and an adsorbed TLR7/8 agonist [141]. The agonist used is an IMQ-like molecule, named IMDG, which significantly enhanced neutralizing antibody responses in animal models, with a distinct Th1 bias, and increased levels of SARS-CoV-2-specific IFN-γ+ CD4+ T lymphocyte response.

A final example of an aluminum-containing combination adjuvant is represented by the formulation of AH with cytosine/guanosine oligodeoxynucleotide (CpG ODN), a TLR9 agonist immunopotentiator already approved for human use [142]. Negatively charged CpG ODNs readily adsorb to AH, although phosphate ions inhibited optimal adsorption in a concentration-dependent manner [143]. Several reports on the AH–CpG ODN combination adjuvant have been performed using different antigens ranging ovalbumin [144], SARS-CoV-2 RBD subunit [145], and hepatitis B surface antigen (HbsAg) [146]. This combination adjuvant will be discussed more in depth in an accompanying review in this journal (Dynavax).

## 8. Conclusions

Aluminum-containing adjuvants have been administered in billions of doses of vaccines over almost 100 years, mostly to children and adolescents. Although they have limitations, such as their relative inability to induce robust cell-mediated (Th1) immune responses and a susceptibility to freezing, their record of safety and tolerability, along with their low cost, continues to make these adjuvants an attractive component of vaccines. As we discussed recently, aluminum adjuvants remain the gold standard against which new and exploratory adjuvants should be evaluated [5]. Recent advances in the biophysical characterization of aluminum-adjuvanted vaccines, including sophisticated high-throughput screening methods, facilitate the development of new vaccine formulations and enable quality control during the manufacturing process. Taking advantage of the high adsorptive capacity of aluminum adjuvants, they will increasingly be used as a platform to develop novel combination adjuvants able to drive the necessary immune responses to specific pathogens and for use in selected human populations. These developments will ensure that aluminum-containing adjuvants remain a mainstay of vaccine formulations for the foreseeable future. 

## Figures and Tables

**Figure 1 pharmaceutics-15-01884-f001:**
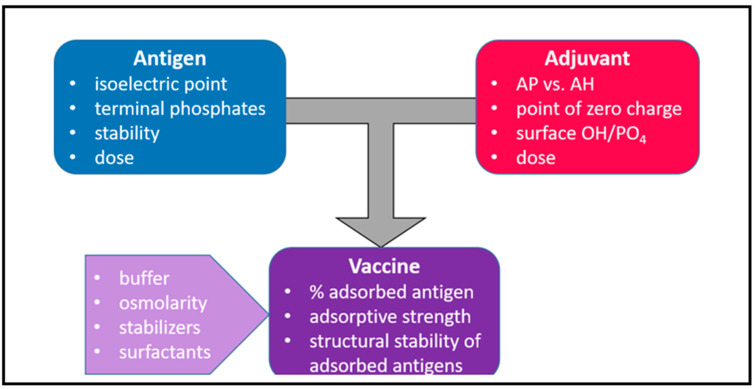
The development of vaccines with aluminum adjuvants requires considerations of the type of aluminum adjuvant and dose as well as a thorough characterization of the antigen. The degree and strength of adsorption and the effect of adsorption of the antigen should be determined. These parameters are influenced by the excipients including the type and osmolarity of the buffer, stabilizers, and surfactants. Adapted from Reference [5].

**Figure 2 pharmaceutics-15-01884-f002:**
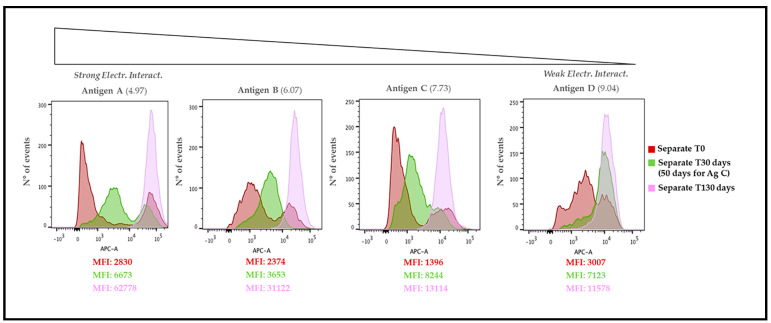
FC analysis of a tetravalent AH-adjuvanted vaccine prepared using a formulation strategy named separate adsorption: with this approach, each protein antigen (Ag) is individually adsorbed onto AH and the four monovalent samples are pooled together later. The four Ag used in this study are representative of different electrostatic strengths according to their isoelectric points (as reported in brackets). Overlay of the histograms at different time points shows the maturation of distribution of each single Ag over the AH particles: from highly polydispersed samples at T0 (dark red), to monodispersed samples after 130 days (light violet). In middle, samples after 30/50 days of incubation (light green) show maturation trend according to each electrostatic strengths: still polydispersed for Ag A, while already similar to sample after 130 days for Ag D. The X-axis indicates fluorescence intensity and the Y-axis number of events. All numbers refer to mean fluorescence intensity (MFI) values. From Reference [87].

**Figure 3 pharmaceutics-15-01884-f003:**
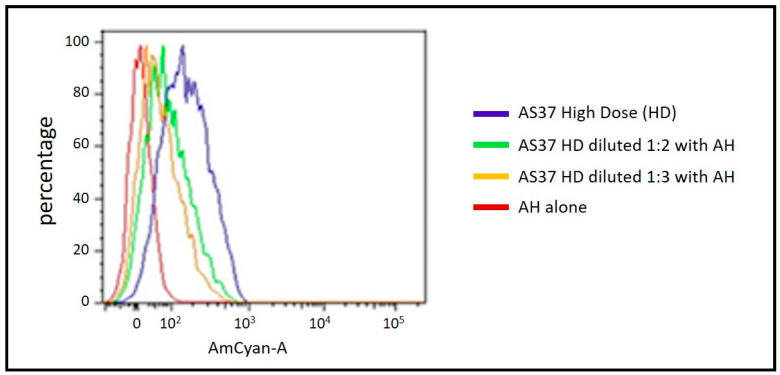
Flow cytometry analysis of Adjuvant System 37 (AS37), which comprises aluminum hydroxide (AH) and an adsorbed TLR7 agonist. The histogram shows how the AH alone (red trace), which displays a negative fluorescence intensity, shows an increase in fluorescent signal after formulation with a high dose (HD) of TLR7a (blue trace): these data represent a direct confirmation of TLR7a adsorption onto AH. AS37 HD has a unique and monodispersed peak, meaning that almost all AH particles deliver a similar amount of agonist. The samples in green and orange instead represent the AS37 HDs that were, respectively, diluted 1:2 and 1:3 with AH alone to reach a lower TLR7 agonist dose whilst keeping the AH amount constant. As a consequence of sample dilution, the fluorescence intensity decreased relative to the TLR7a dose, always maintaining a monodispersed TLR7a distribution over AH particles. X-axis indicates fluorescence intensity and the Y-axis the percentage of the maximum number of events. Adapted from Reference [127].

**Table 1 pharmaceutics-15-01884-t001:** Type of aluminum-based adjuvant and their content in vaccines licensed for human use in the United States (Information accessed on FDA website on 17 January 2023: Vaccines Licensed for Use in the United States|FDA). * Pediatric dose, ° Adult dose.

Vaccine	Approval Date	Tradename	Manufacturer	Aluminum Used	Dose (Al^3+^)
Anthrax	1970	Biothrax	Emergent BioDefense Operations Lansing (Lansing, MI, USA)	AH	0.6 mg
Diphtheria and tetanus	1997	None	Sanofi Pasteur (North York, ON, Canada)	AP	0.33 mg
	2003	TENIVAC	Sanofi Pasteur	AP	0.33 mg
	2018	TDVAX	MassBiologics (Mattapan, MA, USA)	AP	0.53 mg
Diphtheria, tetanus, and acellular pertussis (DTaP)	1997	INFANRIX	GSK (Rockville, MD, USA)	AH	0.5 mg
	2002	DAPTACEL	Sanofi Pasteur	AP	0.33 mg
DTaP, hepatitis B, and poliovirus	2002	PEDIARIX	GSK	AH/AP	0.85mg
DTaP, Haemophilus b, and poliovirus	2008	Pentacel	Sanofi Pasteur	AP	0.33 mg
DTaP, Haemophilus b, hepatitis B, and poliovirus	2018	VAXELIS	MSP Vaccine Company (Swiftwater, PA, USA)	AAHS	0.32 mg
DTaP and poliovirus	2008	KINRIX	GSK	AH	0.5 mg
	2015	Quadracel	Sanofi Pasteur	AP	0.33 mg
Tetanus, reduced diphtheria, and acellular pertussis (TdaP)	2005	ADACEL	Sanofi Pasteur	AP	0.33 mg
	2005	BOOSTRIX	GSK	AH	0.3 mg
Haemophilus B	1989	PedvaxHIB	MSD (Cincinnati, OH, USA)	AAHS	0.23 mg
Hepatitis A	1995	HAVRIX	GSK	AH	0.25 *–0.5 ° mg
	1996	VAQTA	MSD	AAHS	0.23 *–0.45 ° mg
Hepatitis B	1986	RECOMBIVAX HB	MSD	AAHS	0.25 *–0.5 ° mg
	1989	ENGERIX-B	GSK	AH	0.25 *–0.5 ° mg
	2021	PreHevbrio	VBI (Cambridge, MA, USA)	AH	0.5 mg
Hepatitis A and B	2001	Twinrix	GSK	AH/AP	0.45 mg
Human papillomavirus	2006	Gardasil	MSD	AAHS	0.23 mg
	2009	CERVARIX	GSK	AH	0.5 mg (plus 50 µg of MPLA)
	2014	Gardasil 9	MSD	AAHS	0.5 mg
Japanese encephalitis	2009	IXIARO	Valneva (Saint-Herblain, France)	AH	0.25 mg
Meningococcal group B	2014	TRUMENBA	PFIZER (Kalamazoo, MI, USA)	AP	0.25 mg
	2015	BEXSERO	GSK	AH	0.52 mg
Pneumococcal	2010	Prevnar 13	PFIZER	AP	0.13 mg
	2021	PREVNAR 20	PFIZER	AP	0.13 mg
	2021	VAXNEUVANCE	MSD	AP	0.13 mg
Tick-borne encephalitis	2021	TICOVAC	PFIZER	AH	0.18 *–0.35 ° mg

**Table 2 pharmaceutics-15-01884-t002:** List of analytical tools useful for providing different types of information directly on aluminum-adsorbed antigen(s) without any desorption needed. Full names of techniques listed in the table: attenuated total reflectance Fourier transform infrared (ATR-FTIR), near-infrared (NIR), circular dichroism (CD), intrinsic fluorescence (IF), extrinsic fluorescence (EF), differential scanning calorimetry (DSC), nuclear magnetic resonance (NMR), confocal microscopy and high content fluorescence imaging analysis (Imaging), flow cytometry (FC), enzyme-linked immunosorbent assay (ELISA), direct Alhydrogel formulation immunoassay (DAFIA), Raman spectroscopy (Raman), and liquid chromatography–mass spectrometry (LC–MS).

Analytical Tool	Type of Information	Application	Type of Vaccine	References
ATR-FTIR	Secondary protein structure: focus on β-strand conformation	Routinely implemented: challenging data interpretation	Only monovalent	[61,62,63,64,65,66]
NIR	Determination of the adsorbed protein content	Exploratory	Only monovalent	[67]
CD	Secondary protein structure: focus on α-helix conformation	Routinely implemented: challenging data interpretation and execution	Only monovalent	[68,69,70]
IF	Tertiary protein structure: focus on conformational changes; Determination of the adsorbed protein content: useful for inline process monitoring and HTP analysis.	Routinely implemented	Only monovalent	[63,64,65,71,72,73,74]
EF	Tertiary protein structure: focus on conformational changes.	Routinely implemented	Only monovalent	[68,75,76]
DSC	Tertiary protein structure: focus on thermal stability.	Routinely implemented	Only monovalent	[77,78,79]
NMR	Atomic structural analysis: solid state analysis performed on adsorbed antigen.	Exploratory	Only monovalent	[80,81]
Imaging	Determination of the adsorbed protein content; Antigen distribution/orientation on aluminum adjuvants.	Exploratory	Mono- and multivalent	[82,83,84]
FC	Determination of the adsorbed protein content; Antigen distribution on aluminum adjuvants; Focus on freeze/thaw damage.	Exploratory	Mono- and multivalent	[85,86,87,88]
ELISA/DAFIA	Determination of the adsorbed protein content; Useful for QC analysis: focus on antigenicity, potency, and product consistency.	Routinely implemented	Mono- and multivalent	[89,90,91,92,93]
Raman	Structural fingerprint of complex vaccine products: high specificity and sensitivity.	Exploratory	Mono- and multivalent	[94]
LC–MS	Determination of the adsorbed protein content.	Exploratory	Mono- and multivalent	[95]

## Data Availability

Not applicable.

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
