# Peer review of "Aluminum Adjuvants—‘Back to the Future’"

_pharmaceutics, 2023, doi:10.3390/pharmaceutics15071884_

Round 1

Reviewer 1 Report

This is a very nice comprehensive review on the use of "alum" for immunization. The topic is interesting for a broad readership. No critizism or additions from my side.  The manuscript can be recommended for publication in the present state.

Author Response

We thank the reviewer for his/her nice comments on our manuscript

Reviewer 2 Report

The submitted review by O’Hagan et al, provides formulation, characterization, and molecular insights regarding the usage of different aluminum-based adjuvants within vaccine formulations. The manuscript is well-written describing recent approaches and assessments for developing such immuno-modulating formulations. Suggestions and comments are to be addressed prior publication:

1. Authors should provide literature data comparing the application of different vaccine formulation additives (e.g. aluminum versus saponin and others), highlighting the advent of employing which adjuvant.

2. Providing schematic diagrams regarding the formulation steps of each aluminum adjuvants within vaccine formulations would be more relevant and highly attractive for the readers.

3. In Section 3.2. Authors should provide reference(s) for the cited surface charge values.

4. Introducing a schematic diagram at section 4, is highly recommended for highlighting the suggested mechanistic aspects for how aluminum-based adjuvants within vaccine formulations could modulate immune responses.

5. Authors should describe current evidence regarding the use of aluminum adjuvant within multiple associated vaccine proteins. This approach was suggested  beneficial particularly if each could mount different immune responses; humoral and/or cellular (please refer to Evaluation of the Association of Recombinant Proteins NanH and PknG from Corynebacterium pseudotuberculosis Using Different Adjuvants as a Recombinant Vaccine in Mice”, Vaccines 2023, 11(3), 519; https://doi.org/10.3390/vaccines11030519).

6. Limitations as well as challenges regarding the application of aluminum-based adjuvants within vaccine formulations should be thoroughly described.

Minor editing of English language required

Reviewer 3 Report

In the manuscript, authors revisit the current topic of aluminum adjuvants, their types, physiochemical characteristics, biological properties, chemical-analytical characteristics, and the next generations of aluminum-based adjuvants. In particular, the latter part, dedicated to adjuvants combined with aluminum and other proteins or molecules, is very timely, detailed, and robust.

In general, the manuscript reviews almost all essential points related to aluminum adjuvants and is well-structured and well-written. Still, I have several suggestions the authors should take into account and address when they find them relevant before I recommend the editor(s) consider the manuscript's publication.

(i) Firstly, I would recommend adding a paragraph about other types of non-aluminum adjuvants. There is a note that aluminum adjuvants are the oldest ones and most commonly applied in the Introduction on lines 36--39. However, a short paragraph about the alternatives, i.e., oils and organic adjuvants such as saponins or bacterial or other organic products, is welcome, although many of the options for aluminum adjuvants are outdated.

(ii) A paragraph about aluminum adjuvants' biological and clinical safety would work. Some ideas are already included in lines 237--242. Although aluminum salts are generally registered and considered safe, many papers mention possible associations between aluminum adjuvants and potential neurotoxicity and maybe other ways of toxicity, namely,

-- Tomljenovic L, Shaw CA. Aluminum vaccine adjuvants: are they safe? Curr Med Chem. 2011;18(17):2630-7. doi: 10.2174/092986711795933740. PMID: 21568886.

-- Shaw CA, Li D, Tomljenovic L. Are there negative CNS impacts of aluminum adjuvants used in vaccines and immunotherapy? Immunotherapy. 2014;6(10):1055-71. doi: 10.2217/imt.14.81. Erratum in: Immunotherapy. 2019 Apr;11(6):555. PMID: 25428645.

-- Tomljenovic, L. Aluminum and Alzheimer's Disease: After a Century of Controversy, Is there a Plausible Link?. Journal of Alzheimer's Disease. 2010;23(4):567–598. doi: 10.3233/JAD-2010-101494. PMID 21157018.

(iii) Although the manuscript is written in good English, the authors should address some minor language and formatting issues. First, there needs to be an abbreviation explanation in the abstract for TLR on line 19 (I guess for Toll-like receptors), which first occurs on lines 56--57. Article "a" should not be at the end of the previous lines but at the beginning of the following lines (lines 99, 251, 298, 372, etc.). Second, the quality of the raster images listed in the manuscripts is relatively low -- I suggest replacing them with copies of higher resolution. Third, it is a good habit to place an introducing sentence between a section and a subsection title, e.g., between lines 586 and 587 or between 338 and 339, like you did, e.g., between lines 137 and 144.

English language in the manuscript is of mean to high quality. However, minor edits, such as the article "a" position within a line, are required.

Reviewer 4 Report

Overall

The manuscript entitled: "Aluminum Adjuvants-´Back to the future´ submitted by Laera et al is a comprehensive, useful, and well-written review of the properties of aluminum adjuvants from either biological, chemical, or physicochemical points of view.

Major comments

-Lines 176-174, the authors addressed the factors that affect antigens adsorption over aluminum adjuvants, I consider that time of adsorption is another important factor that is missing, either for the initial adsorption or for the stability of the antigen.  This aspect was analyzed in these studies: doi: 10.3390/vaccines11010155, doi: 10.1016/j.vaccine.2020.02.001, doi: 10.1002/jps.23422, and doi: 10.1002/jps.22180

 I strongly suggest this variable be included because, in my personal experience, this simple but important aspect can improve the immunogenicity of an antigen, therefore a kinetic of adsorption is advisable to choose the best time of full adsorption before an immunogenicity protocol is executed.

-Lines 223-224, I disagree with the statement “The full mechanism of action of aluminum adjuvants is still poorly understood,..” since several reports have proven that inflamasomme, specifically NLRP3 has a major role in the mechanisms of action. A review (doi: 10.3390/vaccines8030554) containing various papers that supports this, and an original paper (doi: 10.1016/j.vaccine.2016.04.081) are references that need to be checked and probably cited.

Minor comments

-Line 488, since ELISA is an immunological method instead of a physicochemical one, I suggest that the 5.3.2. section be moved after Raman spectroscopy and Lc-MS section.

-Line 580, Limulus is an specie so that it must be written with an initial capital letter.

-Line 638, the correct spelling of the virus is SARS-CoV-2.

-Line 650, the Greek character alpha is missing in the cytokine TNF and a hyphen is missing in IFN-g.

Round 2

Reviewer 2 Report

The authors adequately responded to almost all suggestions. However, I still consider providing schematic diagrams regarding aluminium adjuvant formulation as well as highlighted mechanistic aspects of aluminium adjuvant-associated immune response modulations. Both would be relevant and highly attractive for the readers and signify this review as being more comprehensive, rather that just provide referral to published article.  

Minor editing of English language can be proceeded

Author Response

We have added a figure that outlines the considerations that go into the formulation of vaccines with aluminum adjuvants (Figure 1 – page 3).